# Research on a New Soundscape Evaluation Method Suitable for Scenic Areas

**Jing Liu [1,\*], Ziyan Dan [1] and Zengfeng Yan [2]**

1   School of Art, Northwest University, Xi'an 710060, China; danziyan@stumail.nwu.edu.cn
2   School of Architecture, Xi'an University of Architecture and Technology, Xi'an 710055, China; yanzengfeng@xauat.edu.cn
\*   Correspondence: 20200023@nwu.edu.cn

**Abstract:** Existing studies have focused mainly on the environmental quality of scenic spots, such as sufficient oxygen content in the air and a high concentration of negative oxygen ions. The perceptions of soundscape in scenic areas are generally good, but there are few reports on the quantitative evaluation of soundscape quality in scenic areas. In this study, we analysed existing methods for evaluating the soundscape of a landscape, evaluated the soundscape comfort of scenic spots, analysed and refined the natural environmental factors affecting the soundscape, and proposed for the first time to use physical environmental indicators such as the air temperature difference, relative humidity, natural illuminance ratio and wind speed as environmental evaluation variables. A quantitative method was used to calculate the soundscape comfort index (SSI) of the landscape. The physical environmental indicators related to famous scenic spots in China, namely, Qingcheng mountain field testing and a subjective soundscape of tourist satisfaction survey, were used to calculate the corresponding soundscape comfort index values, and a quantitative analysis of soundscape comfort and differences in temperature, relative humidity, the illumination ratio, and the correlation between the equivalent sound level A was performed. The measured values of the temperature difference and light ratio were significantly correlated with the soundscape comfort index. The distribution of sound landscape comfort was given by a GIS map, and soundscape comfort was evaluated scientifically. The correlations between soundscape comfort and landscape patch number (PN), landscape patch density (PD), diversity index (Shannon), and landscape shape index (LSI) were quantitatively analysed, which confirmed that the perception of soundscape comfort was affected by landscape space to different degrees. This study has scientific significance and application value for the soundscape evaluation of scenic areas and has significance for soundscape evaluation and design strategies for urban landscapes.

**Keywords:** soundscape quality; scenic areas; soundscape comfort index; total environment; landscape; urban parks





## 1. Introduction

The concept of a soundscape, as defined by the ISO in 2014, refers to the sound environment that a person can imagine, feel, or understand in a certain scenario [1]. Scholars have described the aspects of soundscapes as the acoustic environment, sound wave environment, sound environment, sonic environment, audio environment, natural acoustic environment, environmental sound, sound natural environment, natural condition environment, quiet area, area with good environmental noise quality, area with high acoustic quality, urban soundscape, overall ambient sound environment, overall soundscape, and acoustic soundscape [2]. Obviously, the concept of soundscape is an interdisciplinary field involving multiple domains, such as physical acoustics, environmental sciences, architecture, and ecology [3]. Soundscape comfort is a comprehensive evaluation of people's physical and psychological satisfaction with the objective environment of a sound

landscape. Therefore, the evaluation of soundscape comfort is a comprehensive method involving various fields and aspects.

Existing studies on soundscape assessment include evaluation methods and various factors that affect the results [4–9]. In the latest research on soundscape assessment, the exploration of the systematic relationship between urban environmental factors and soundscape assessment includes the correlation between the natural environment and the pleasure dimension, as well as the influence of natural and urban conditions and urban management on the physical environment [10,11]. However, people's evaluation of the soundscape is influenced by the presence or absence of accompaniment. Some studies have proposed the importance of accompaniment in soundscape evaluation and concluded that the sound of human activities can promote the acceptance of social sounds among peers, while people who are alone may prefer privacy and quiet [12]. For urban soundscape evaluation, the survey methods for different soundscape evaluations in ISO 12913-2 [1] include qualitative and quantitative methods. The research shows that the quantitative data scheme is suitable for a large group and derives a generalised model, while the qualitative data scheme is suitable for a small group or in-depth analysis of some sites [13]. Urban parks are public spaces with abundant soundscape resources. The evaluation and analysis of the soundscape of park roads show that there is a certain relationship between soundscape satisfaction and sound source structure elements, sound source preference, sound pressure level, landscape configuration and other factors [14–17]. From the perspective of urban soundscape management, some scholars have concluded that the subjective and acoustic classification of soundscape is the first step of evaluation, and a classification model has been proposed as a tool for the comprehensive evaluation of urban soundscape. This model aims to automatically classify urban soundscapes based on underlying acoustic and perceptual criteria [18]. In summary, in terms of various environmental regions, due to different regional forms, the evaluation of a soundscape by a single type of element is often insufficient [19]. Therefore, the impact of multiple elements in the landscape environment on the evaluation results of the soundscape is very important. Soundscape quality assessment involves many disciplines: acoustics, physiology, sociology, psychology, anthropology and statistics. Soundscape quality assessment is a fuzzy evaluation method [20,21].

It is well known that the subjective perception of soundscape in scenic areas is generally good, but quantitative evaluation of the relationship between subjective perception and environmental factors of soundscape in scenic areas is rare [22,23]. The soundscape evaluation of scenic areas is worthy of scientific research [24–28]. At the same time, the study of soundscape evaluation in scenic areas also provides insights for the evaluation and construction methods of urban landscapes. In this study, it is necessary to construct a soundscape comfort evaluation method that considers the physical indices of the landscape environment (temperature, humidity, light, wind speed, etc.), which is lacking in current soundscape evaluation research.

This study aims to address three problems: 1. Soundscape comfort evaluation is often limited to the quantitative analysis of sound elements, and there is a lack of analysis on whether it is more relevant to the changes in environmental elements caused by the characteristics of underlying landscape elements. 2. Soundscape comfort and the underlying landscape structure are unified; the two should not be separated, and their mutual relationships should not be ignored. 3. The mechanism of exploration of superior soundscapes in scenic areas may help to improve the quality of the acoustic environment of urban landscapes and parks.

## 2. Method

### 2.1. Research Area

Existing studies mainly focus on the environmental quality of scenic spots, such as sufficient oxygen content in the air and a high concentration of negative oxygen ions. The subjective perception of soundscape in scenic areas is generally good, but there are few reports on the quantitative evaluation of soundscape in scenic areas. The research

object is Qingcheng Mountain, an ecological mountainous scenic area with rich natural and artificial landscape resources in Sichuan Province, China, as illustrated in Figure 1. This is a representative case of research on methods for evaluating soundscape quality in scenic areas.

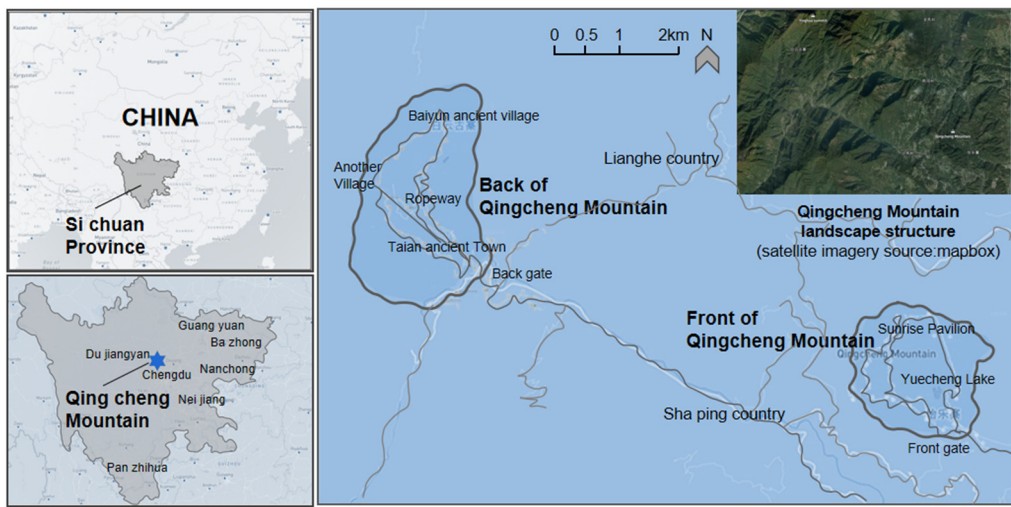

**Figure 1.** Location map of Qingcheng Mountain.

As a world cultural heritage site, Qingcheng Mountain is the holy land of Taoism Quanzhen in the world, one of the four famous Taoist mountains in China. It is located southwest of Dujiangyan city, Chengdu city, Sichuan Province, China, 68 km to the east of Chengdu city, and 10 km southwest of Dujiangyan hydraulic engineering [29]. Qingcheng Mountain has beautiful scenery and numerous cultural relics. The natural landscape is spectacular, with beautiful water and a secluded forest. It is cold in the winter and cool in the summer in the mountains. It is often said that Qingcheng Mountain, which has a secluded forest, is number one in the world [30]. The secluded feature of Qingcheng Mountain, the seminatural scenic area, with a combination of humanity and nature landscapes, creates a comfortable soundscape environment with "quiet" characteristics, which is representative of the soundscape evaluation of the scenic area.

Qingcheng Mountain is divided into a front mountain and a back mountain, and the front mountain is the main part of the Qingcheng Mountain scenic area, with a length of approximately 15 square kilometres and numerous cultural relics. The back mountain covers an area of 100 square kilometres. Qingcheng Mountain, located in moist evergreen forest in mid-subtropical areas, is rich in all kinds of plants. The total forest area of the Mountain Qingcheng Scenic Area is 2350 hectares, which is covered with two ancient trees beside temples, colourful forest clusters, and large, magnificent areas of pure forest plantations. It has a subtropical humid monsoon climate with four distinctive seasons, is moist and rainy, and is hot in the summer but not cold in the winter. The annual average temperature is 15 °C, with a high temperature of 32 °C and a low temperature of −8 °C in recent years. The annual average air humidity is 86%, and there is an annual average sunlight of 102.4 h, accounting for 23% of the duration of possible sunshine. Typically, the winds are calm. Considering the application of the soundscape assessment method in China, the validation research was tested in the autumn and winter.

*2.2. Soundscape Data Measurement*

For analysis, the area was divided into 38 parts in the Qingcheng Front Range and Back Range. These areas were surveyed from 9:00 20 October 2017–20:00 22 October 2017–9:00 23 December–20:00 25 December, with Mountain Qingcheng as the research object, the landscape node in the area centre as the test starting point, and scenic areas at equidistance as the test points. To guarantee that the testing was completed within half an

hour, four teams were used, and all tests started at the same time. In total, 38 test points were measured by four teams. Spot monitoring of the equivalent continuous sound level A, temperature, relative humidity and natural illumination were performed. The weather conditions during the measurements were cloudy to sunny on 20 October, 12.7–17.7 °C; sunny to overcast on 21 October, 13.8–18.7 °C; clear to overcast on 22 October, 12.9–17.4 °C; clear for three days; light rain to cloudy on 23 December, 5–11.1 °C; cloudy to overcast on 21 December, 6.9–11.1 °C; overcast to light rain on 22 December, 5.8–18.4 °C; and overcast for three days. The average wind velocity in the measured areas was 0.24 m/s. A sound level metre (HS6280D), thermohygrometer (L93-2L) and illuminometer (TES1330A) were used as monitoring instruments.

The SPL, temperature, humidity and illumination were measured for 38 scenic areas. The data are presented in Table 1 and Figures 2 and 3.

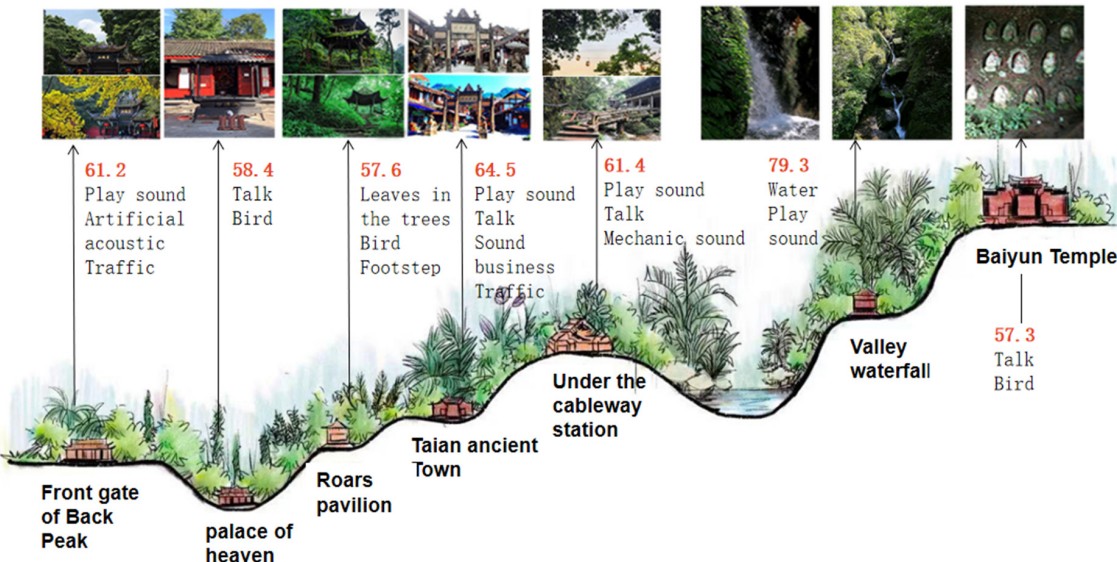

**Figure 2.** Schematic of the scenic spots in front of the Qingcheng Mountain.

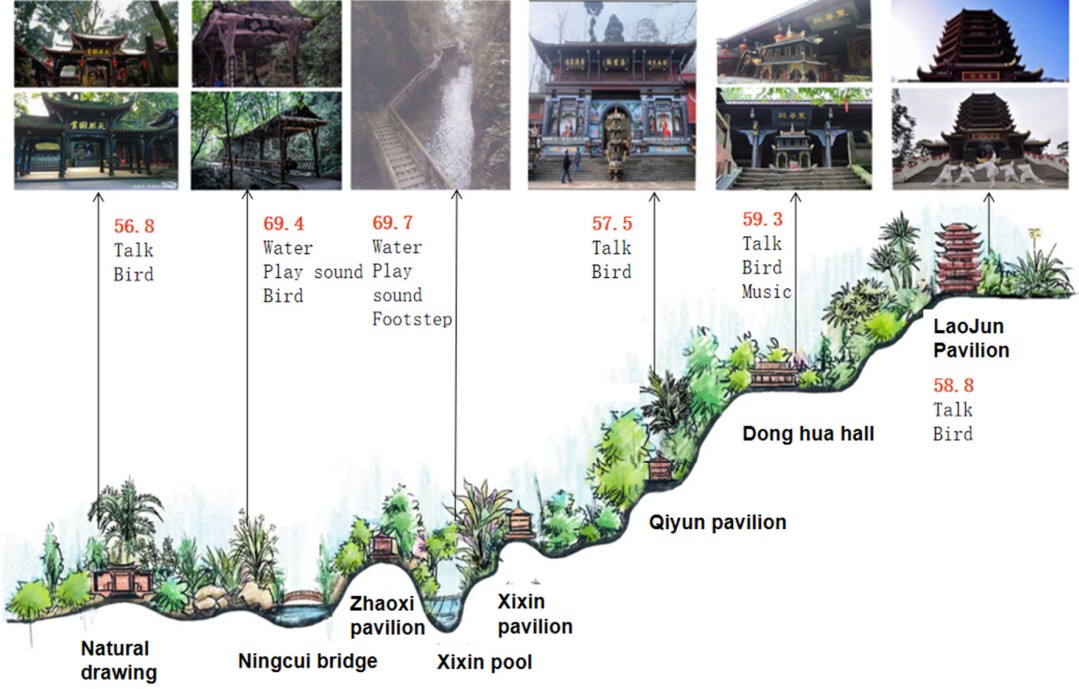

**Figure 3.** Schematic of the scenic spots in the back of Qingcheng Mountain.

**Table 1.** Measured natural variables at different scenic areas in Qingcheng Mountain.

| Scenic Area | Equivalent SPL (dBA) | Temp (°C) | RH (%) | Illuminance (lux) | V (m/s) | Scenic Area | Equivalent SPL (dBA) | Temp (°C) | RH (%) | Illuminance (lux) | V (m/s) |
|---|---|---|---|---|---|---|---|---|---|---|---|
| Front gate | 66.4 | 18.7 | 80 | 1430 | 1.6 | Front gate of Back Peak | 61.2 | 9 | 73 | 950 | 1.5 |
| Natural pavilion | 44.6 | 17.7 | 75 | 130 | 1.4 | Bai Zhang Bridge | 59.7 | 6.9 | 72 | 373 | 1.6 |
| Yile Nest | 61.2 | 14.4 | 87 | 430 | 1.7 | Mandarin Duck Island | 66.4 | 8.4 | 71 | 59 | 1.8 |
| Natural drawing | 56.8 | 14.2 | 92 | 2070 | 1.8 | Valley waterfall | 79.3 | 8.3 | 79 | 165 | 1.9 |
| Yuecheng Lake | 47.7 | 15.5 | 86 | 2420 | 1.1 | Huxiao Pavilion | 67.7 | 8.6 | 77 | 586 | 1.7 |
| Taoist Master's Cave | 49.2 | 13.4 | 84 | 3470 | 1.2 | Taian ancient Town | 64.5 | 8.5 | 72 | 756 | 1.7 |
| Shramana Pavilion | 50.3 | 12.7 | 87 | 655 | 1.6 | Wind Pavilion | 40.3 | 5 | 86 | 2560 | 2.1 |
| Fifth Cave | 50.3 | 13.3 | 82 | 434 | 1.5 | Another Village | 54.9 | 6.2 | 83 | 6250 | 2.0 |
| Shadow Pavilion | 50.4 | 14.7 | 80 | 265 | 1.4 | Hidden Dragon Gorge plank | 75.0 | 8.6 | 73 | 4780 | 1.5 |
| Quanzhen Taoism Temple | 55.4 | 16 | 69 | 5140 | 2.0 | Fog Spring | 75.9 | 8.9 | 71 | 4130 | 1.4 |
| Sunny Cave | 42.1 | 13.8 | 89 | 867 | 1.4 | Haiman Pavilion | 68.9 | 10.1 | 63 | 3180 | 1.4 |
| Ancestral Hall | 46.6 | 13.7 | 86 | 117 | 1.1 | Shulao Pavilion | 69.2 | 11.1 | 60 | 3290 | 1.1 |
| Fangning Bridge | 46.1 | 14.2 | 87 | 120 | 1.2 | Homesick Pavilion | 46.1 | 10.8 | 48 | >20,000 | 1.8 |
| Pen-throwing Slot | 36.3 | 14.2 | 95 | 291 | 1.2 | Thatched Cottage | 48.6 | 18.4 | 32 | >20,000 | 1.9 |
| LaoJun Pavilion | 60.2 | 12.6 | 98 | 4010 | 1.6 | Ksitigarbha Cave | 52.0 | 6.1 | 62 | 2180 | 2.0 |
| Donghua Hall | 52.0 | 14.8 | 99 | 1045 | 1.8 | Reclining monk Cave | 52.8 | 6.5 | 62 | 2650 | 1.9 |
| ShengdengPavilion | 48.7 | 13.1 | 90 | 655 | 1.4 | Nine monks Cave | 61.4 | 6.7 | 62 | 5440 | 1.6 |
| ShangQing Palace | 64.4 | 13 | 94 | 623 | 1.5 | Xiong-er Pavilion | 39.4 | 5.8 | 64 | 3220 | 1.8 |
| Sunrise Pavilion | 50.7 | 13.2 | 90 | 685 | 1.3 | Tongtian Cave | 49.0 | 6.8 | 62 | 6940 | 1.4 |

A soundscape subjective satisfaction survey was also used to assess the comfort level and subjective satisfaction of 38 parts.

The sound sources with higher favourability in the subjective evaluation of the front and back mountains are shown in Figure 4. Scenic areas with these rich sound source types have better soundscape evaluation results. Visitors reported good soundscape comfort in Shangqing Hall, Yuecheng Lake, Guanri Pavilion, Shengdeng Pavilion, and Chaoyang Cave. Areas with low soundscapes were Laojunge Pavilion, Xixin Kiosk, Tongtian Cave, and Guanzhi Pavillion. Visitors found these places noisy. No feelings about the soundscape were reported for Cliffside Spring in Hidden Glen, Huxiao Pavilion, Yuanyang Island, or Natural Pavilion.

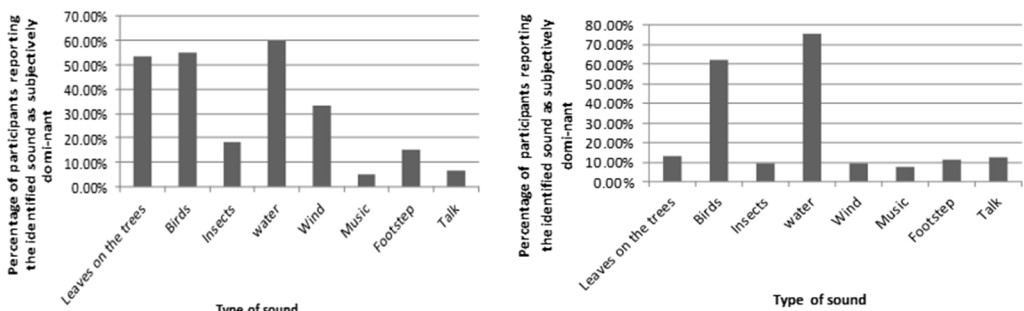

**Figure 4.** The percentage of subjective evaluation of the dominant sound source in front (**left**) and back (**right**) of the Qingcheng Mountain.

### 2.3. Establishment of the Soundscape Comfort Index

Studies have shown that human physiological function and mental health are comprehensively influenced by temperature, humidity, illumination, wind speed, and other natural environmental factors [31]. The combined action of these factors can be used as thermal environmental assessment indices to establish the comfortable degree index of the human body (CIHB) [32,33]. This study used these four natural environmental physical indices to construct an SSI assessment system to construct evaluation indices that can reflect the diversity of the effects of natural environmental factors in scenic areas on the quality of the soundscape and human physiological and psychological feelings [31].

In this study, the degree of comfort of a sound landscape can be evaluated according to the human thermal comfort index [32,33]. The evaluation indices of sound landscape comfort included temperature deviation, relative humidity, illumination ratio, and wind speed as the basic factors [32]. Research has suggested that there is a linear relationship between the comfort index of the human body and the degree of comfort of sound [33]. According to the thermal comfort theory, the evaluation index is between −3 and +3. Considering the human body sound perception with a unidirectional asymmetrical feature, which means excessive noise and quite silence cause discomfort either [32], we propose that the comfort index of a soundscape should be between 0 and 3. The calculation of the comfort index of the sound landscape is proposed as follows:

$$SSI = 0.36 \times \triangle T - 0.11 \times (3.6 \times \triangle T - 26) \times (1 - RH) - 0.064 \times (V) \, 1/2 + 0.64 + 0.32 \times (i/i0)$$

SSI is the proposed soundscape comfort index and has a value between 0 and 3, a dimensionless quantity. The soundscape comfort value corresponds to subject satisfaction. The corresponding relationships are shown in Table 2.

**Table 2.** SSI assessment values and subjective feelings.

| Level | Index Range | Soundscape Feeling Degree |
|---|---|---|
| 1 | 0~1 | Most people have no strong soundscape impression; |
| 2 | 1~2 | Few people have a certain sound feeling but the soundscape aesthetic feeling is not strong. |
| 3 | 2~2.5 | Comfortable sound feeling, most people have a good soundscape impression. |
| 4 | 2.5~3 | Comfortable soundscape feeling. Left with a good hearing impression except for visual feeling. |
| 5 | ≥3 | Not a comfortable soundscape experience. Most people perceive the area as noisy. |

$\triangle$T is the temperature difference between the test points and the landscape environment's exterior background (in °C);

RH is the relative humidity, and the unit is %.

i/i0 is the ratio between the test points and the natural illumination of the landscape exterior background and is a dimensionless quantity. Due to test errors, the number is 1 when i is larger than i0.

V is the wind speed, in m/s.

The above physical environmental indices can be used for the quantitative assessment of a landscape area. These indices can be used to calculate soundscape comfort and to analyse the influence of landscape design on environmental factors. Under their combined action, the subjective experience of the soundscape of the human body is formed quantitatively.

*2.4. Statistical Analysis*

The nonparametric (K independent samples) Kruskal-Wallis test was used to collect the perceptual differences of each type of sound from 37 measuring points. The relationship between human physiological sensation and overall sound level in different soundscape environments was studied at two levels. Spearman correlation analysis was performed for the main sound sources. The percentage of the main sound sources in the overall soundscape was calculated for each sampling point (spatial scale) and each sampling period (temporal scale). A correlation analysis between the subjective evaluation and objective test data of the soundscape comfort at each measuring point was carried out. Correlation analysis between the main soundscape elements and landscape indices was also carried out. All the above statistical analyses were carried out in SPSS.

**3. Results**

In this study, four physical indices, namely, temperature, humidity, light, and wind speed, were used to evaluate the comfort of scenic areas with a variety of natural environmental factors. These factors can reflect the quality of scenic areas with respect to soundscape quality and can be used to evaluate human physiological and psychological perceptions. Studies have shown that these four physical indices and the relationships among sounds are objective and scientific.

*3.1. SSI Computation Values*

According to the above assessment values, SSI indices were calculated for the 38 scenic areas. A mountain Qingcheng soundscape comfort level distribution diagram was constructed as shown in Figures 5 and 6.

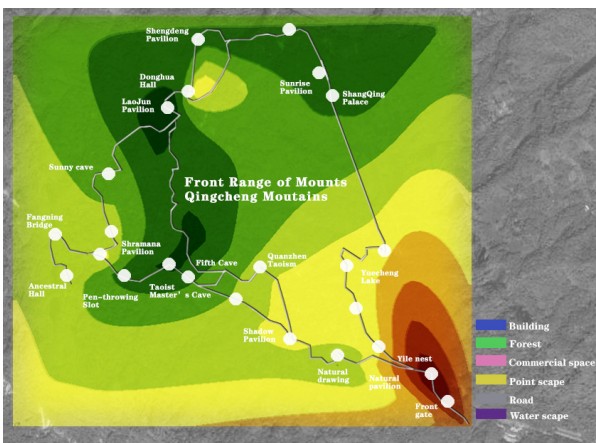

**Figure 5.** Mountain Qingcheng Front Range soundscape comfort distribution diagram.

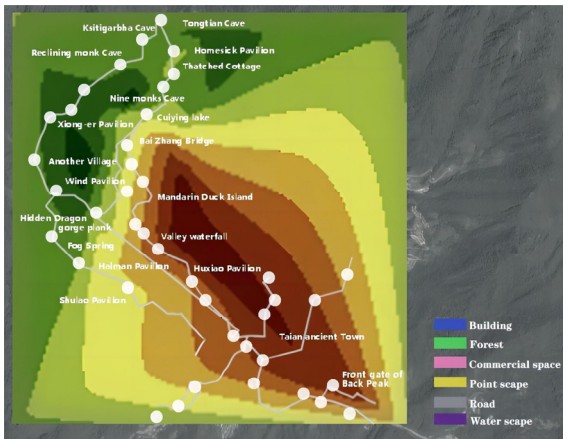

**Figure 6.** Mountain Qingcheng Back Range soundscape comfort distribution diagram.

The landscape comfort calculation method proposed in this research has a good correlation with tourists' subjective feelings on Mount Qingcheng. Together with a GIS distribution diagram, it can intuitively express the soundscape comfort of a scenic area.

### 3.2. SPSS Results

SPSS (V19.0) was used for correlation analysis of the soundscape comfort assessment and temperature difference, relative humidity, illumination ratio, and equivalent sound level A for each location. The analysis results are presented in Table 3:

**Table 3.** SSI assessment values and objective parameter correlation analysis results (Correlation coefficient/significance level).

| SSI/Temperature Difference | SSI/Relevant Humidity | SSI/Illumination Ratio | SSI/Equivalent Sound Level A |
|---|---|---|---|
| 0.926/0.000 ** | 0.117/0.643 | 0.344/0.162 * | 0.383/0.117 * |

Notes: * and ** in the table refer to the significance level, * indicates $p < 0.05$, and ** indicates $p < 0.01$.

The calculation results show a significant correlation between the actual measured temperature difference, illumination ratio and the SSI assessment index. Additionally, a correlation was observed between the relative humidity and the assessment index. Studies have shown that the SSI calculation method can objectively calculate the SSI. The combination of this objective calculation with subjective assessment provides an assessment of soundscape comfort.

### 3.3. Landscape Space Analysis

A detailed analysis of the composition of the soundscape in the research area revealed that there is a close relationship between the soundscape and the underlying landscape structure. According to the previous analysis of the contribution of the dominant sound, the composition of the soundscape differed between sampling sites, as shown in Figures 7 and 8.

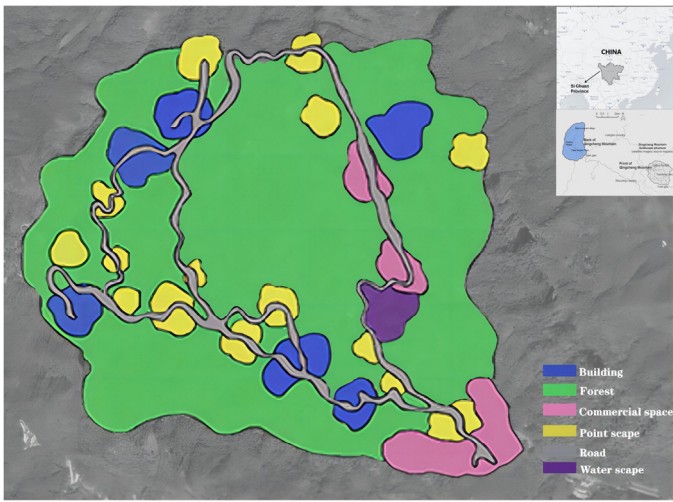

**Figure 7.** Landscape space division in front of Qingcheng Mountain.

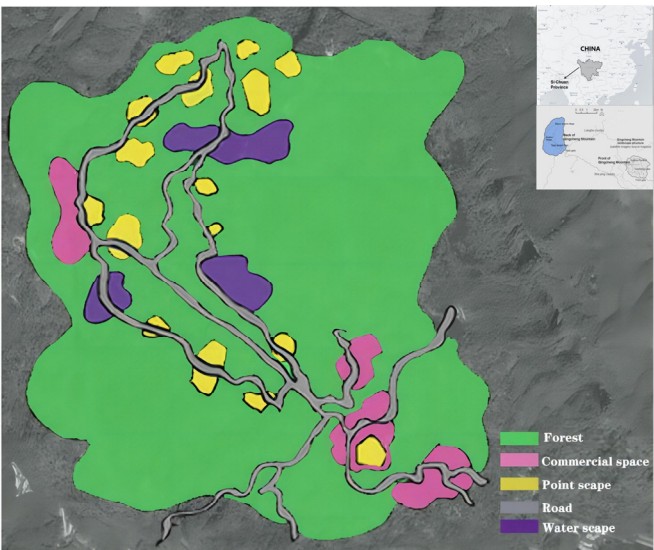

**Figure 8.** Landscape space division in the back of Qingcheng Mountain.

In commercial spaces and building areas, the sounds of human activities are mainly adult voices, children's voices and footsteps. In general, natural sound is considered the dominant sound at the 38 sampling points on Qingcheng Mount. This is especially true at landscape points (water, forests) away from main roads and where fewer tourists gather. The front gate, front gate of the back peak, Taian ancient town, and Yuecheng Lake ropeway are located in commercial spaces. Therefore, artificial voices (47.2%) were dominant. In Yuecheng Lake (68%), Bai Zhang Bridge (54%), Valley Waterfall (56.5%), Hidden Dragon Gorge Plank (49.8%), and Fog Spring (76.2%), water is the main landscape element, and the sound of water flow is dominant. Taoist Master's Cave, Fifth Cave, Quanzhen Taoism Temple, LaoJun Pavilion Donghua Hall, Shang Qing Palace and other enclosed forms of buildings and landscape nodes. The surrounding environment is surrounded by buildings

or dense vegetation, and the noise level is relatively low. Compared with commercial space, the composition of sound sources increases the number of birdsong and natural sounds (leaves, wind, etc.). Similarly, natural pavilions, Shramana pavilions, Shadow Pavilions, Shengdeng Pavilions, Xiong-er Pavilions, Huxiao Pavilions and other construction spaces lack the enclosure of buildings, and semiclosed or open spatial forms decrease the number of people. Therefore, part of the sound source composition reduces the participation of talking sounds, children's sounds and playing sounds. Natural sources are more prominent. This kind of point space has relatively high landscape and soundscape comfort in mountains with dense vegetation. The results show that the composition of the soundscape is significantly correlated with the underlying landscape characteristics [34].

In Qincheng Mountain scenic areas with superior natural conditions, more noise can be generated by vehicles, people and other factors in commercial spaces, which confirms that traffic sounds are the main source of noise in the environment. However, when entering the interior of a scenic area, vegetation density can significantly reduce the perceived loudness of noise. Areas with dense vegetation can often become high-quality habitats for sound organisms, such as birds and insects. Landscape patch number (PN), landscape patch density (PD), diversity index (Shannon) and landscape shape index (LSI) showed that [3] since the shape complexity and morphological richness of surface patches can promote the formation of biodiversity, this may be because fragmented landscapes can also provide a wider ecological niche for a wider variety of bird species [35]. At the sound composition level of each landscape type, the dominant sound showed a significant relationship with at least two landscape indices (Table 3). High-quality sound sources such as natural sounds, bird songs and insect sounds are highly perceived in environments with dense vegetation, such as water and forests [36]. Artificial sound sources such as conversation sounds and play sounds are more prominent in buildings and point spaces. Traffic sounds (TSFs) are more likely to be perceived in commercial spaces. Birdsong (BS) is the most significant component of biological sound in the research area, is more likely to be perceived in areas close to point spaces and forests, which reflects the influence of landscape patch density (PD) and the landscape shape index (LSI) on biological sounds.

Brumm's previous research revealed that organisms that produce sounds (such as birds) may sing louder to counteract the masking effect of traffic noise [37], so the perception of bird song is also prominent in building spaces within mountains. In Qincheng Mountain scenic areas with high vegetation density and water space dominated by water sounds, tree sounds are closely related to vegetation density, which is also affected by the patch richness density (PRD). The vegetation density shown by the NDVI has the greatest impact on urban soundscape perception because it affects the perception of all three major sounds and most major subclasses of sounds [38]. Buildings and roads are the main sound sources, and their density and location particularly affect the perception of foreground traffic sounds and birdsong. Landscape fragmentation measured by the landscape shape index (LSI) affects the perception of many sounds, especially foreground traffic sounds, while landscape heterogeneity reflected by landscape patch density (PD) has no close relationship with soundscape perception in this research [39–41]. In conclusion, different landscape features influence human perceptions of certain sounds to different degrees, as shown in Figure 4. Table 4 shows the correlation between the soundscape comfort at each landscape point and the four landscape indices.

**Table 4.** Soundscape comfort evaluation values and objective parameter correlation analysis results (correlation coefficient/significance level).

| SSI/PN | SSI/PD | SSI/Shannon | SSI/LSI |
|---|---|---|---|
| 0.6250.000 * | 0.487/0.002 ** | 0.6860.001 ** | 0.629/0.001 ** |

Note: * and ** in the table represent the significance levels. ** indicates statistical significance at the 0.01 level.

The results in Table 4 fully demonstrate that different landscape features affect landscape comfort to different degrees.

## 4. Discussion

An application of the proposed SSI method to a Qingcheng mountain park is conducted in this research. The results provide useful information for improving the quality of soundscapes, especially for improving the performance of natural landscapes. Notably, for a systematic and comprehensive project of park soundscape design, the analysis of other soundscape-related attributes may be needed. Although only natural landscape elements are surveyed and analysed in this study, the practicality and effectiveness of the proposed SSI method for improving the soundscape of urban parks are well demonstrated. The simplification and efficiency of the method make it easy to implement and therefore suitable for the quality measurement of all the key soundscape-related attributes. There are also three points that need to be discussed.

### 4.1. Relationships between Four Environmental Elements and Sound Perception

#### 4.1.1. Temperature–Sound Relation

The sound velocity will change with the air temperature and pressure. It decreases at lower temperatures, and sound is transmitted faster in the air. A study of the influence of temperature, noise, and musical sounds on the CIHB concluded that people have stronger resistance to exterior cold stimulation and accept a high SPL outside [33], but a hot environment inside will cause significant discomfort and seem noisy. In landscape design, many factors affect people's perceptions [34]. Therefore, changing the temperature before and after assessing the SSI can provide insight into the influence of the landscape, water, and underlying surface design on heat island effect remission.

#### 4.1.2. Humidity–Sound Relationship

The sound velocity is related to the air humidity. Sound transmission occurs much faster in humid air than in dry air. Studies have shown that the SSI is negatively related to the SPL within the humidity range of 18.85~63.77% but is positively related to the SPL within the humidity range of 63.77~73.75%. Thus, humidity can greatly affect SSI assessment. Air relative humidity can reflect the influence of greening, water and underlying surface design on regulating the water vapour content in the air in landscape design and can be used to measure the degree of change in landscape comfort.

#### 4.1.3. Illumination–Sound Relationship

Well-lit environments are generally preferred. Although people may stop under a shady tree, most choose to stay in the sun when the temperature and humidity are near a certain value. The duration of time spent in the sun can be controlled and selected by people according to their own feelings. Surveys show that people prefer light and shadow environments produced by uniformly spaced plants and trees. Subjective survey data show that at the demarcation point of the light environment assessment on comfort and discomfort, i.e., the demarcation point of SSI, the comfort zone corresponds to the comfort zone, and the discomfort zone corresponds to the discomfort zone. Both of these trends are positively correlated.

In the SSI assessment, the illumination ratio, or the ratio between test points and the natural illuminance of the landscape exterior background, was used. The dimensionless quantity shows the influence of the landscape and underlying surface design on the visibility of changing skies.

#### 4.1.4. Wind Speed–Sound Relationship

Sound velocity changes in the air under the influence of wind. It increases under favourable wind conditions and decreases under unfavourable wind conditions. The sound velocity in the air is approximately 340 m/s, and the wind speed is usually several metres to more than ten metres, so the wind speed has only a small influence on the sound velocity. Wind speed can reflect changes in the landscape and underlying surface design to affect air flow in a landscape area and influence people's feelings towards the landscape.

### 4.2. The Total Environment and the Sound Environment Information Representation Factor Acquisition

The elements of the natural environment are constantly changing with changes in weather, time and people. Human comfort is basically determined by human activities and the perception of the natural environment. Therefore, several aspects should be considered when using the method in this research to obtain information on each element of the soundscape comfort evaluation index. First, the elements of the natural environment should be recorded in a way that is easy to understand and easy to use. In this study, the values of various environmental elements of each landscape point were recorded in groups, and it should be noted that the same period should be maintained. All the members of the research group have a certain degree of recognition of the experiment from their own or related majors to improve the professionalism of the recording method and subjective questionnaire survey. The results of the tests were averaged over a specific period as perceived by the members of the research group. At the same time, people's subjective perceptions of various sound sources should be considered, not just the equivalent sound level. In addition, the appropriate spatial scale based on the perceptions of the members of the research group will help to understand the relationship between soundscape comfort and spatial landscape patterns, as well as the various natural environmental elements that affect the perception of soundscape comfort.

In this study, we selected scenic areas with superior environments to analyse the composition of their landscape space and how to influence and build high-quality soundscape environments. This is a meaningful and valuable model for evaluating natural and urban environments, but it is important to consider a longer span and a wider range of seasonal variations. Long-term research has focused on collecting information on environmental indicators related to soundscape comfort.

### 4.3. The Use of the Soundscape Comfort Evaluation Method

The study of soundscapes cannot be understood in isolation from the environmental background. The landscape creates conditions for the construction of a soundscape [6,39]. The evaluation of the comfort level of soundscape in the environment and the establishment of a standard evaluation method for scenic areas can provide a basis for the creation of high-quality soundscape because soundscape with more natural sound is preferable [42,43]. The landscape index is a scientific tool that quantifies spatial landscape patterns and is used to assess and evaluate landscape patterns [44,45]. However, the application of these indices to soundscape data has been neglected [46]. In our research, the human thermal comfort index is taken as the basis for establishing soundscape comfort, and the four basic elements of the environment are included in the category of the soundscape comfort evaluation method. The perception of human thermal comfort affects the perception of the acoustic environment, not just the value of the environmental sound level. Research has also shown that the composition of the landscape determines the composition of the soundscape, that is, which sounds are more prominent; spatial landscape structure affects sound transmission and thus affects the soundscape pattern. In the process of establishing the evaluation method, scenic areas with superior natural environments were used as objects, and the correlation between the soundscape comfort index and spatial landscape pattern index was verified by regional and point-by-point Spearman correlation analysis [47]. Therefore, the landscape index is regarded as the link between soundscape comfort and landscape distribution [48]. The close relationship between soundscape comfort and the spatial landscape index indicates that soundscape comfort evaluation provides a basis for the construction of soundscape environments and the practice of landscape management in urban parks. For example, the landscape shape index (LSI) is positively correlated with biodiversity, and landscape fragmentation caused by the complexity of land use patch shape can also promote the perception of biodiversity and provide a wider ecological niche for a wider variety of birds [34], which directly affects the perception of biological

sounds. This indicates the need to use complex and diverse landscape types to improve the soundscape environment [49–51].

It is also interesting that some potential effects of landscape-soundscape interactions can be seen from this study. For example, a strong (Spearman's correlation coefficient, 0.686) and a significant ($p$ value < 0.01) positive correlation is found between the perceived satisfaction of the sound environment and that of the Shannon index, as shown in Table 4. This is because visual quality is one of the most important factors that affects people's perceptions of a park's total environment. This result demonstrated that the qualities of the soundscape and landscape have significant effects on each other. These results imply that audio-visual interactions can have significant effects on soundscape assessment and are worth considering in future studies.

The soundscape comfort evaluation involves a series of indices of physical elements in the environment. The soundscape comfort evaluation method obtained in this study can be used to evaluate the soundscape comfort of a natural environment. In future research, to verify the relationship between the soundscape comfort value and subjective soundscape perception, additional types of high-quality soundscape space and landscape types in different seasons should be investigated. Future research should consider the spatiotemporal scale of the landscape in the urban context.

## 5. Conclusions

In this study, a highly accurate multiperformance soundscape comfort prediction model was established using various environmental parameters, and model interpretation and parameter analysis were conducted. The main conclusions are as follows:

(1) Human body comfort should not be ignored in landscape construction and promotion. The research results show that, in addition to the subjective evaluation methods, the evaluation of the environmental factors that affect human body physiological function and psychological feelings related to physical factors such as temperature, humidity, light, and wind speed should be considered when evaluating soundscape comfort more objectively and scientifically.

(2) Due to changes in seasons and time, scenic areas experience a variety of feelings of soundscape, and the shape of the landscape space in the environment affects its physical factors. The temperature, humidity and light availability are more comfortable in areas with high landscape diversity and high plant density than in areas with external commercial and building spaces. At the same time, the landscape diversity index is high, which brings more biological sounds, especially bird song, which contributes more prominently to the overall soundscape.

(3) This study confirms that spatial variation in soundscape patterns is closely related to the underlying landscape characteristics; seasonal variation is related to the composition of plant landscape patterns and biological sounds; and temporal variation affects the perception of artificial sounds in the environment. Soundscape comfort is affected closely by landscape patterns. The landscape patch density (PN), landscape patch density (PD), diversity index (Shannon), and landscape shape index (LSI) will have different degrees of impact on the soundscape comfort.

(4) These spatial relationships between soundscape and landscape indicate that to improve the comfort of soundscape in an urban environment, more natural sounds and biological sounds should be added, high-quality sound sources should be established through landscape spatial layout, and uncomfortable sound sources should be filtered. Especially in urban environments where there are many discordant sound sources, such as artificial sound, traffic sound and mechanical sounds, it is particularly helpful to use the proposed soundscape comfort evaluation method to improve the quality of the acoustic environment of urban landscapes and parks, which is the original intention of the novel findings of this study.

**Author Contributions:** All the authors contributed to the study conception and design. Material preparation, data collection and analysis were performed by J.L., Z.D. and Z.Y. The first draft of the manuscript was written by J.L., and all the authors commented on previous versions of the manuscript. All the authors have read and approved the final manuscript. All authors have read and agreed to the published version of the manuscript.

**Funding:** This work was supported by the National Natural Science Foundation of China (No: 51978554, No: 52278127).

**Institutional Review Board Statement:** Not applicable.

**Informed Consent Statement:** Not applicable.

**Data Availability Statement:** Data are contained within the article.

**Acknowledgments:** The authors would like to acknowledge the graduate students at Xi'an University of Architecture and Technology who helped collect data from Qingcheng Mountain Park.

**Conflicts of Interest:** The authors declare no conflict of interest.

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
