# Peer review of "Research on a New Soundscape Evaluation Method Suitable for Scenic Areas"

_sustainability, doi:10.3390/su16093707_

Round 1
Reviewer 1 Report
Comments and Suggestions for Authors
Line 8
It seems that “SSssd” represents the ”soundscape comfort index” and “soundscape subjective satisfaction and comfort index” at the same time in the Abstract. Please make it correct.
Besides, the abbreviation needs to appear only once when you mention the corresponding words first. It does not need to be repeated.
Line 9
What is the meaning of “perceptions of soundscape managers”? Is it an occupation?
Line 15
It is unclear here “to use … as variables” for what?
Line 33
The introduction should give more literature reviews related to the study topic. For instance, the introduction has no content regarding soundscape studies in scenic areas. Instead, a lot of effort was put into discussing urban soundscape evaluations.
Line 77
Subjective evaluations can also be quantitative. They are not “black and white”.
Line 90
This section is more like the introduction for the study background instead of the description of methods, which can be integrated into Section 1.
Line 111
Please try to reduce the contents of this section and only leave those that are basic and relevant to the present study.
Line 172
There should be a figure (map) to showcase where these sampling points are located.
Line 187
There are some display problems at the head of this table.
Line 206
Which research has suggested this? Please add the reference.
Line 208-209
Why does the thermal comfort index between -3 and +3 serve as the reason for the comfort index between 0 and 3?
Line 211
It is unclear how this formula comes here based on the above contents.
Line 230-235
These contents seem to be a part of the Results Section.
Line 250-262
These are also like the study results.
Line 264-269
This segment looks like the beginning of the Discussion or Conclusion Section.
Line 270
These results should be indicated according to corresponding tables or figures.
Line 309
In this section, the authors should display more about their study results instead of citing the findings of previous studies.
Line 351
It should be Table 4 here.
Comments on the Quality of English LanguageThe english language of this paper can be more concise and concrete and avoid redundancies.
Reviewer 2 Report
Comments and Suggestions for Authors
1. The reference list lacks recent literature from the past two years, and the reference formatting is inconsistent throughout.
2. Prior to submission, the author should carefully review the formatting of all figures and tables, as some appear to have display issues.
3. The SSssd metric mentioned in the text - is this being introduced for the first time? If so, a more precise description is needed. If not, appropriate references should be provided.
4. In section 2.4 where computational formulas are presented, the author should provide more detailed explanations of the evaluation metrics used and the underlying theoretical foundations, supported by relevant references.
5. The innovative contributions of this work should be stated more explicitly by the author.
6. The abbreviation "SSssd" is unclear and should be clarified or replaced with a more descriptive term.
Reviewer 3 Report
Comments and Suggestions for Authors
Paper with Title: "Research on a new soundscape evaluation method suitable for scenic areas" is presented to be published in Environmental Sustainability and Applications in Special Issue Environmental Noise Assessment and Analysis for a Sustainable Environment.
This work mainly focuses on the environmental quality of scenic spots, such as sufficient oxygen content in the air and a high concentration of negative oxygen ions. The perceptions of soundscape managers in scenic areas are generally good, but there are few reports on the quantitative evaluation of soundscape quality in scenic areas. In that study, authors analyzed existing methods for evaluating the soundscape of a landscape, evaluated the soundscape comfort of scenic spots, analyzed and refined the natural environmental factors affecting the soundscape, and proposed using environmental physical indicators such as the air temperature difference, relative humidity, natural illuminance ratio and wind speed as variables. The measured temperature difference and light ratio values were significantly correlated with the soundscape comfort index (SSssd). A GIS map gave the distribution of sound landscape comfort, and soundscape comfort was evaluated. The correlations between soundscape comfort and landscape patch number (PN), landscape patch density (PD), the diversity index (Shannon), and landscape shape index (LSI) were quantitatively analyzed, which confirmed that the perception of soundscape comfort was affected by landscape space to different degrees. I agree with the authors That the study has scientific significance and application value for the soundscape evaluation of scenic areas and has importance for soundscape evaluation and design methods for urban landscapes.
This work in the journal issues and discusses a sound topic.
I think this work is fit to publish after
- Moderate English editing
- Add aim of the work in a separate paragraph at the end of the introduction
- Figures 5,6,7 and 8 need to increase in size and resolution.
Comments on the Quality of English LanguageEnglish Moderate editing of English language required
